# Emission Enhancement of Ge/Si Quantum Dots in Hybrid Structures with Subwavelength Lattice of Al Nanodisks

**DOI:** 10.3390/nano13172422

**Published:** 2023-08-25

**Authors:** Vladimir A. Zinovyev, Zhanna V. Smagina, Aigul F. Zinovieva, Aleksei A. Bloshkin, Anatoly V. Dvurechenskii, Ekaterina E. Rodyakina, Margarita V. Stepikhova, Artem V. Peretokin, Alexey V. Novikov

**Affiliations:** 1Rzhanov Institute of Semiconductor Physics, Siberian Branch of Russian Academy of Sciences, 630090 Novosibirsk, Russia; smagina@isp.nsc.ru (Z.V.S.); aigul@isp.nsc.ru (A.F.Z.); bloshkin@isp.nsc.ru (A.A.B.); dvurech@isp.nsc.ru (A.V.D.); rodyakina@isp.nsc.ru (E.E.R.); 2Department of Physics, Novosibirsk State University, 630090 Novosibirsk, Russia; 3Institute for Physics of Microstructures of the Russian Academy of Sciences, 603950 Nizhny Novgorod, Russia; mst@ipmras.ru (M.V.S.); aperetokin@ipmras.ru (A.V.P.); anov@ipmras.ru (A.V.N.)

**Keywords:** photoluminescence, Al nanoparticles, plasmonic, photonic waveguide modes, bound states in the continuum (BICs)

## Abstract

The effects of resonance interaction of plasmonic and photonic modes in hybrid metal-dielectric structures with square Al nanodisk lattices coupled with a Si waveguide layer were investigated using micro-photoluminescence (micro-PL) spectroscopy. As radiation sources, GeSi quantum dots were embedded in the waveguide. A set of narrow PL peaks superimposed on the broad bands were observed in the range of quantum dot emissions. At optimal parameters of Al nanodisks lattices, almost one order increasing of PL intensity was obtained. The experimental PL spectra are in good agreement with results of theoretical calculations. The realization of high-quality bound states in the continuum was confirmed by a comparative analysis of the experimental spectra and theoretical dispersion dependences. The results demonstrated the perspectives of these type structures for a flat band realization and supporting the slow light.

## 1. Introduction

Currently, there is a significant progress in the application of photonic crystals (PhCs) to create efficient emitters working in a highly demanded telecommunication range [1,2,3,4,5,6]. Usually quantum dots (QDs) are used as radiation sources embedded in PhCs. Luminescence enhancement in such structures is due to an increase in the probability of emission transitions, when QDs are placed in the maximum amplitude of the electric field inside the PhCs resonator [3]. One of the suitable versions of radiation sources is SiGe QDs. This type of QDs is compatible with Si-based technology, and its emission bandwidth coincides with desired near infrared range [4,5,6]. Many scientific groups are now investigating the possibility of amplifying the QDs emission by embedding them in PhCs. An additional possibility of emission enhancement is the use of the principles of plasmonics [7]. Creating the arrays of metallic islands, on the surface of a heterostructure with embedded QDs, can allow to obtain amplification of QDs emission by several times [8,9,10]. The combination of two approaches and the development of metal–dielectric PhCs structures representing the periodical structures of metallic islands on the surface of waveguide dielectric layer with embedded QDs can be considered as a promising method to create effective emitters with additional control of plasmonic and photonic resonances. A great emission enhancement can be achieved if such structures can support the bound states in the continuum (BICs). These states have very high-quality factors and already found applications in various areas, such as non-linear enhancement, coherent light generation, sensors, filters, and so on [11]. There are several types of BICs; the first type is formed due to the symmetry incompatibility with the free space propagating waves [12,13,14]. These BICs exist as long as the symmetry is preserved. The second type includes interference-based BICs, when the destructive interference of resonances leads to the formation of dark states [15,16,17]. The BIC resulting from the interference of two resonances that belong to the same resonator is known as Friedrich–Wintgen BIC (FW–BIC) [18]. These type of BICs can be obtained by tuning the parameters of the system that can lead to a complete elimination of the radiation. This provides high flexibility in the engineering of the mode dispersion, allowing high control on their frequency positions, angular span, and linewidth [17]. 

The most intensive studies of BICs have been conducted for PhCs structures [11]. For metal–dielectric structures there is also a lot of research in this direction [19,20,21], but so far, no high-quality resonances typical of PhCs have been obtained. Due to losses, the broad resonances with low Q-factors are usually observed for metallic nanostructures. The high quality factor is an intrinsic feature of special type of states, bound states in continuum [22]. Usually, these states are observed in dielectric structures, and their realization in metallic systems is a very hard problem. Recently, the hybrid system with metallic grating coupled with a dielectric optical waveguide was investigated theoretically [17], and the possibility of realization of the bound states in the continuum (BICs) with diverging Q-factors was demonstrated. Moreover, in work [23] the plasmonic BIC states are realized in all metallic systems with a Q-factor exceeding 60. Also, the authors of [23] theoretically showed that, with optimal parameters of metallic grating, one can obtain Q-factor exceeding 200. In the present work, we study the effects of the interaction of GeSi quantum dots emitters with hybrid plasmonic-photonic waveguide modes, in metal–dielectric PhC structures, with regular subwavelength Al nanodisk lattice, and find the optimal parameters that allow the realization of BICs. 

## 2. Materials and Methods

To create a waveguide structure with embedded SiGe QDs as emitters, a silicon-on-insulator (SOI) substrate with a 90 nm thick top Si layer and a 3 μm thick SiO_2_ layer was used. At the first step, a 280 nm thick Si buffer layer was grown by molecular beam epitaxy at 500 °C. At the next step, five layers of GeSi QDs separated by 25 nm thick silicon interlayers were grown at 600 °C. Each QD layer was created by deposition of 7 monolayers of Ge. At the final stage, the structure was covered with a 10 nm thick Si layer. The total thickness of the multilayer structure was 480 nm. Then, using lift-off techniques, square lattices of Al disks with different disk diameters and periods were formed on the surface of grown structure. The height of the disks was 50 nm; diameters were 150, 200, and 250 nm. For each diameter, square lattices with periods of 400 nm, 440 nm, 460 nm, 480 nm, 500 nm, 540 nm were created. A schematic representation and scanning electron microscopy image of the metal–dielectric structure with embedded GeSi QDs are shown in Figure 1. The luminescent properties of the structures were studied by micro-photoluminescence (micro-PL) method with high spatial and spectral resolutions. The micro-PL signal was excited by a continuous laser at the wavelength of 532 nm. The laser radiation was focused on a spot of ~10 μm with a Mitutoyo M Plan APO 10× objective (numerical aperture NA = 0.26). Measurements were carried out in the normal incidence geometry of the exciting and detected beams. The PL signal collection angle was 2θ = 30°. The micro-PL signal was detected by a cooled Ge detector using a Bruker IFS 125HR high-resolution Fourier spectrometer. The measurements were carried out at a temperature of 300 K. To analyze the obtained experimental results, we have simulated the emission spectra of the studied metal–dielectric structures. The near-field components distributions are calculated with the 3D finite element frequency domain method [24] with commercial software COMSOL Multiphysics. 

## 3. Results and Discussion

The micro-PL study shows that the creation of periodic arrays of Al disks on the surface of Si waveguide layer with embedded GeSi QDs leads to PL enhancement at certain wavelengths (Figure 2). The micro-PL spectra of the structure demonstrate the presence of narrow peaks (with quality factor ~100) in the emission region of GeSi QDs. For example, the P_1_ (~1.3 μm) and P_2_ (~1.43 μm) PL peaks have quality factors of 110 and 90, respectively. The best gain results are obtained with the following hybrid structure parameters: Al disk diameter *D* = 250 nm, lattice period *a* = 540 nm. In this case, the enhancement of the PL signal intensity by almost one order of magnitude (spectrum with *a* = 540 nm in the left panel of Figure 2), compared to the structure without regular Al nanodisk lattice (spectrum “outside”), was observed. 

It is an interesting result that the position of broad PL peak at 1.43 μm practically does not depend on the period of the Al disk lattice. Only at period 540 nm, its intensity dramatically increases and the narrowing of the peak P_2_ is observed. This result can be interpreted as an appearance of a high quality peak, which is superimposed on the broad peak. In contrast, the position of this narrow PL peak is sensitive to a change in the lattice period. It shifts to a short wavelength region losing its intensity as the lattice period decreases. 

There are also several small asymmetric narrow peaks in the spectrum. Their positions also depend on the lattice period and shift to the higher frequencies with period decreasing. The spectral positions of main narrow peaks P_1_ and P_2_ are indicated for convenience by arrows in Figure 2.

To understand the nature of observed peaks we perform simulations for different model structures, from the simple to the complex one. Firstly, we study the waveguide modes for the SOI structure with Si layer of 480 nm thickness, lying on the buried 3 μm silicon oxide layer. The calculated reflectance spectrum shows the series of deeps related to excitation of waveguide modes [25]. The presence of these modes modifies the wavelength distribution of QD light leading to an appearance of the minimum at ≈1.47 μm in the PL spectrum (see gray curves in Figure 2 and Figure 3). 

Such a picture was already observed in QDs structures grown on SOI substrates [26] and was explained by a reduced intensity of the vacuum field fluctuations [27] and, correspondingly, by a lower spontaneous QDs emission rate at this wavelength. 

Then we calculate the plasmon resonances for the case of periodical square lattice of Al disks. The geometry and size of Al disks are taken according to experiment (*D* = 250 nm, *h* = 50 nm). Since we deal with PL measurements, we calculate the *up*-emission intensity. Point dipole oscillating in plane of the structure is placed under the edge of Al disks at the depth 10 nm. According to recent results [9] this position corresponds to the highest electric field intensity at a plasmonic resonance. At the first step, we model a square lattice of Al disks on a semi-infinite Si substrate. The periodicity of the structure leads to a modification of the plasmon resonance peak, it becomes narrower and more intensive as compared with the case of an isolated Al nanodisk (Figure 4). The position of the main resonance peak *λ*_10_ is defined by the period of square lattice *a*, *λ*_10_/*n* = 2*π*/*k* = *a* [28], where *n* is the refractive index of Si. For larger periods it shifts to a longer wavelength region. For a case *a* = 540 nm the main plasmon peak related to dipole resonance should be found at *λ*_10_ = 1.89 μm. This is outside of the emission region of GeSi QDs. As we show below, the use of Si waveguide layer on SiO_2_ shifts the plasmon peaks to desired range of QD emission.

At the next step, we have complicated the calculation model by introducing a buried silicon oxide layer under a Si waveguide layer. The thickness of the Si layer and the silicon oxide layer were chosen in accordance with the experimental structure. We calculate the *up*-emission intensity for the point dipole oriented along *x*- or *y*-direction (tangentially or normally to the edge of the disk, respectively) placed under the edge of the Al disks. In these two cases, we obtain a series of peaks, each of them is a result of the interaction of plasmonic and photonic modes. 

At certain wavelengths, the emitted QDs light scattering on the metallic grating is trapped by the waveguide and can be manyfold enhanced by metallic particles. In the calculated spectra there are very narrow peaks and relatively broad bands. The dependence of the spectrum on the radiation exit angle α shows that the main narrow peaks at λ ≈ 1.3 μm and λ ≈ 1.43 μm do not manifest themselves at zero angle (in the direction normal to the surface). At slightest deviation from the normal, these peaks appear in pairs with satellites. For example, the mode at 1.425 μm has the satellite one at 1.455 μm (Figure 3, left panel); the peak at 1.312 μm appears simultaneously with peak at 1.333 μm (Figure 3, right panel). One of the peaks in the appeared pairs has a greater intensity, and the second has a lower one (see Figure 3, red lines). This picture is typical of wave coupling in resonant grating waveguide structures [29,30]. The peak near λ ≈ 1.3 μm is more pronounced in the *Y*-dipole spectrum, while the peak near λ ≈ 1.43 arises in the spectrum of the *X*-dipole. The calculated near field distributions show that these modes have very symmetrical configurations.

The narrow peak at λ ≈ 1.425 μm (Figure 3, left panel) corresponds to a high symmetry mode that is a superposition of two waveguide modes propagating along *x*- and *y*-direction, not under the center of the disk, but along the line passing through the edge of the disk (see Figure 5). Due to this peculiarity, the metallic losses are relatively low, and the corresponding PL peak has small width, unusual for plasmonic structures. A far field radiation of this mode in the normal direction (the radiation exit angle α = 0°) is suppressed due to high symmetry of mode [14,31]. At the deviation from zero angle the symmetry is lowered, and the mode begins to radiate.

It should be noted that, according to calculations, the additional narrow PL peaks should be present in the spectrum, the peak at 1.742 μm and its satellite at 1.775 μm (Figure 3). However, they are out of range of SiGe QDs emission and are not observed in the experiment. The modes at 1.425 μm and 1.742 μm are the same types of waveguide modes, differing only in the number of half-waves that fits the height of the waveguide (see Appendix A). These modes are excited mainly by *X*-dipole oscillating tangentially to the disk edge, while the mode at 1.312 μm is excited by *Y*-dipole (or *Z*-dipole) oscillating along the normal to the edge (or to the surface) of the disk. The calculation results show that *Y*-dipoles more effectively excite the plasmon resonances and provide the presence of the broad peaks in the PL spectra.

In the calculated spectra of *Y*-dipole there are broad bands with peaks at 1.531 μm and 1.634 μm (Figure 3, right panel). These peaks correspond to the waveguide modes with very high field concentration near the edges of Al disks (Figure 6, propagating along a line equidistant from the disks in *x*-direction (for peak at 1.634 μm) and in *y*-direction (for peak at 1.531 μm). The peak at 1.397 μm corresponds to the waveguide mode propagating right under the center disk line. Namely, this peak is responsible for broad background of the experimental peak near 1.4 μm. The strong difference in the width of narrow PL peak at λ ≈ 1.3 µm, and broad background PL peak at λ ≈ 1.4 μm, is caused by the fact that the first peak corresponds to a localized state with low metallic losses (see Figure 7), and the second peak corresponds to a mode with plasmonic wave propagating under the disk line (Figure 6, top panels).

According to calculation results, the narrow peaks corresponding to the high symmetry modes do not radiate at zero angle, i.e., have the attributes of the symmetry-protected BICs [6,15,32,33]. To demonstrate this, we calculate the up-emission spectrum as a function of the radiation exit angle (a band diagram), that is equal to the dependence on the wave vector *k_‖_*, for the parameters of the structure corresponding to the best experimental result, *a* = 540 nm and *D* = 250 nm. Figure 8 demonstrates the dispersion dependence for the emission of *X*-dipole (Figure 8c) and *Y*-dipole (Figure 8f). *Y-* and *Z*-dipoles have the same composition of spectrum, but the *Z*-dipole has more intensive emission. It is clearly seen that *X*- and *Y*(*Z*)-dipoles have a different set of modes in the emission spectra. Since we have different dipole types in experimental structure, then the measured PL spectra are the superposition of spectra of *X*-, *Y*- and *Z*-dipoles. The calculated results confirm the presence of BICs at λ ≈ 1.3 μm and λ ≈ 1.43 μm demonstrating the suppression of the emission in Γ-point.

It seems to us that the narrow peaks are defined mainly by the interference of QD emission light on the square lattice of Al disks, which in this case can be considered as photonic crystal nodes. To verify this assumption, we calculate the band diagram of a high-index dielectric grating on a top of semi-infinite Si substrate (all-dielectric case). We keep the same geometrical parameters of grating and take refractive index *n* = 4.25 (Ge disks). In the considered range of wavelengths, the extinction coefficient of Ge is negligibly small [34], then there are practically no absorption losses, and plasmonic part is absent in the spectrum. The calculation results (Figure 8b,e) show that the main peculiarities are already apparent in the case of the dielectric lattice; there are modes corresponding to PL peaks at λ ≈ 1.43 μm in spectra of X-dipole and at λ ≈ 1.3 μm in spectra of *Y*(*Z*)—dipole. It is interesting that at these wavelengths the flat bands are realized. This result was confirmed by the experimental dependence of PL spectrum on the collection angle. In the experiments, the position of PL peaks at λ ≈ 1.43 μm and its width does not change with the increasing collection angle (Appendix A). Recently, it was shown that such a band dispersion can be formed in the systems with vertical symmetry breaking. Moreover, the authors [35] demonstrated that by finely tuning the filling of the top layer of photonic structure (changing the parameters of grating) one can obtain the transformation from a conventional quadratic shape to a Dirac dispersion, a flat band dispersion, and a multivalley one. Certainly, our case differs from [35], where one-dimensional grating was considered, but the main defining factor (vertical symmetry breaking) is the same.

To specify the effect of interaction with plasmonic modes, we calculate the band diagrams for the structure with a square lattice of Al disks on semi-infinite Si substrate (plasmonic structure). Figure 8a,d show the corresponding band diagrams with plasmonic modes only. In Appendix A we present more extended information about plasmonic mode dispersion (Appendix A). The comparison of the results obtained for all-plasmonic and all-dielectric structures shows that some plasmonic modes are very close in *k*-space to photonic modes and this results in a very strong interaction. Due to this interaction, selected modes demonstrate the avoided crossing behavior. For mode at λ ≈ 1.43 μm, this leads to an appearance of additional prohibition for the light emission at *k_‖_* ≠ 0 (at *α* ≈ 14° in Figure 8c and Appendix A). Such a picture can be fully explained by formation of FW state [18]. Two resonances pass each other as a function of a continuous parameter, the two channels interfere, resulting in an avoided crossing of their resonances. In theory, one of the channels vanishes entirely and becomes a BIC with an infinite quality factor. In real experimental system with losses, the BIC would collapse to a Fano resonance with a finite lifetime, i.e., become a quasi-BIC [31,36]. In the Appendix A, the nature of the mode before and after the anticrossing point is demonstrated by plotting the electric field distribution. At *α* ≈ 6° the field configuration is typical of plasmonic mode, while at *α* ≈ 20° it becomes like that for waveguide photonic mode (Appendix A). These results confirm the strong coupling of modes at corresponded *k_‖_* [17].

The calculated dependence on the lattice period (Figure 9) shows that the decrease in the period leads to a shift of peaks to the short wavelength range, as it usually occurs for photonic crystals. In the experiments, we also observe this shift, though the shifted peaks lose quite a lot in intensity because they go out of the emission region of quantum dots. For the structures with periods *a* = 460 nm, *a* = 480 nm, *a* = 500 nm in the spectra of *Y*-dipole there is an intensive peak (λ ≈ 1.48 μm for *a* = 460 nm, λ ≈ 1.54 μm for *a* = 480 nm, λ ≈ 1.6 μm for *a* = 500 nm) in the range of QDs emission (~1.3 μm−1.5 μm). But in the experiments these peaks are very weak. We explain this discrepancy by the fact that this mode is found in the tail of the distribution of the QD emissions.

## 4. Conclusions

We demonstrate the possibility of the realization of bound states in the continuum in the metal–dielectric structures with square lattice of Al disks coupled to a Si waveguide layer. The proposed structures can support two different types of BICs due to the symmetry incompatibility with far field radiation and destructive interference of resonances. We find the optimal parameters at which an exclusive set of effects, including the formation of localized states, strong coupling with large band splitting, and slow light, can be observed.

## Figures and Tables

**Figure 1 nanomaterials-13-02422-f001:**
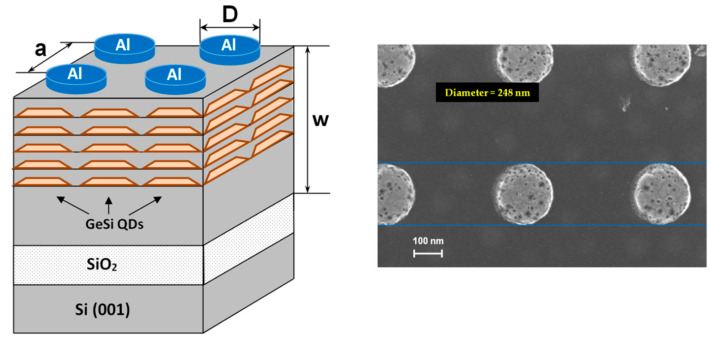
Left—A schematic sketch of the structure under study. Right—A scanning electron microscopy image of the sample with a square lattice of Al disks. Here *a* is a period of Al disk lattice, *D* is a diameter of Al disk, *w* is a thickness of waveguide layer.

**Figure 2 nanomaterials-13-02422-f002:**
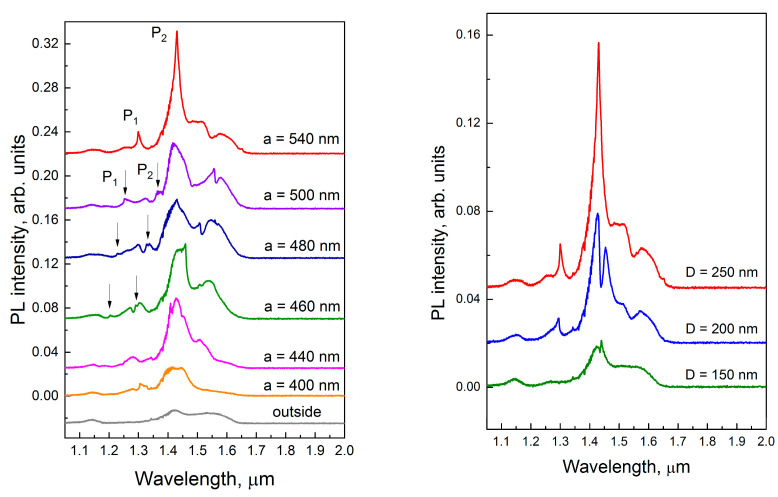
Micro-PL spectra measured at room temperature on the samples with a square lattice of Al nanodisks on the top of a waveguide structure with embedded SiGe QDs. The left panel demonstrates the dependence on the lattice period *a* at fixed disk diameter *D* = 250 nm. The right panel demonstrates the dependence on the diameter of the disk at fixed period *a* = 540 nm. The ‘‘outside” spectrum was measured on the non-processed area, without Al disks. The arrows in the left panel indicate the spectral positions of the P_1_ and P_2_ peaks. Excitation laser wavelength 532 nm, power 20 mW.

**Figure 3 nanomaterials-13-02422-f003:**
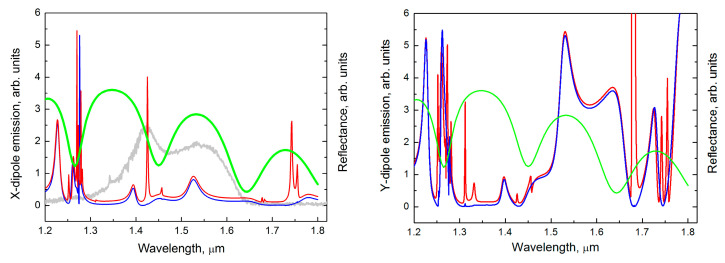
The emission spectra of the model structure with a square lattice of Al nanodisks created on the top SOI structure at radiation exit angle α = 0° (blue line) and at α = 1° (red line). For comparison, the calculated reflectance spectra of the SOI structure with a 480 nm Si waveguide layer lying on a 3 μm buried SiO_2_ layer (green line) is shown. The diameter of the disk *D* = 250 nm, the lattice period *a* = 540 nm. The emission spectra are calculated for the point dipole oriented along *x*-direction (left panel) and *y*-direction (right panel) placed under the Al disk edge at a depth of 10 nm. In the left panel it is also shown the experimental spectrum (gray curve) of the non-processed area of the sample without Al disks.

**Figure 4 nanomaterials-13-02422-f004:**
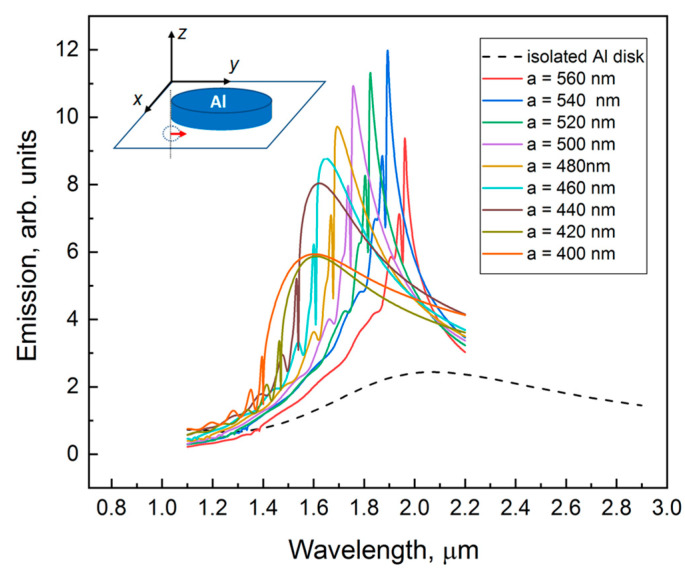
The *up*-emission intensity calculated for the case of a square lattice of Al disk on a semi-infinite Si substrate at radiation exit angle α = 0°. The height of Al disk is 50 nm, the diameter is 250 nm. Point dipole oriented along *y*-direction (*Y*-dipole) is positioned under the edge of Al disks at a 10 nm distance from the surface as shown in the inset. The dashed curve shows the spectral dependence of the *up*-emission intensity calculated for the case of an isolated Al disk on a semi-infinite Si substrate.

**Figure 5 nanomaterials-13-02422-f005:**
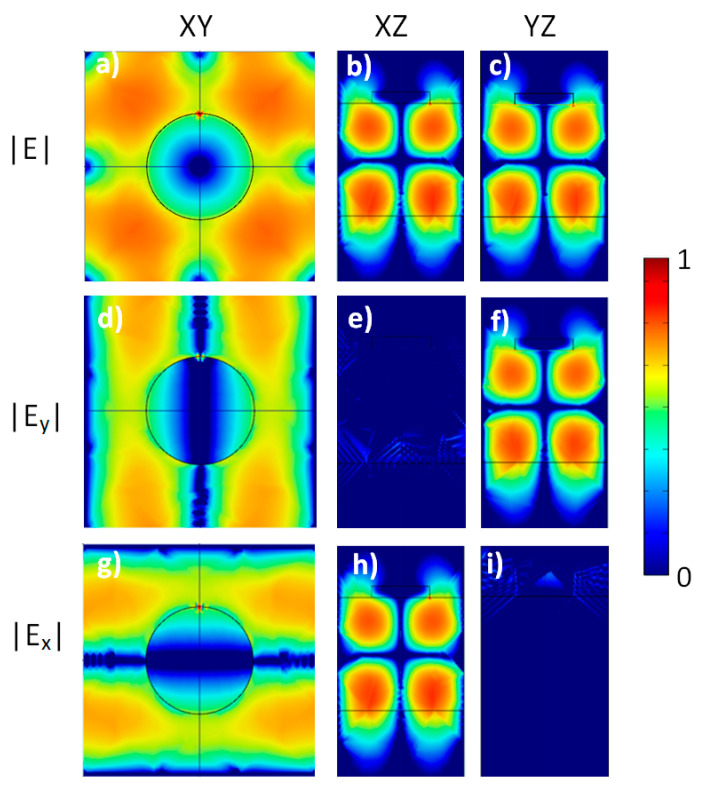
Near field distributions calculated at λ = 1.425 μm for the model structure that is an analog of the experimental SOI structure with a square lattice of Al disks coupled to a Si waveguide layer (*D* = 250 nm and *a* = 540 nm). XY—sections (**a**,**d**,**g**) are taken at the interface air/Si. XZ—sections (**b**,**e**,**h**) and YZ—sections (**c**,**f**,**i**) cross the center of Al disks. Point dipole oriented along *x*-direction (*X*-dipole) is placed under the edge of Al disks at the depth 10 nm. Here, *z*-component of electric field is not shown, because |E_z_| = 0. The calculation was performed at radiation exit angle α = 0°. The color scale is shown in the right.

**Figure 6 nanomaterials-13-02422-f006:**
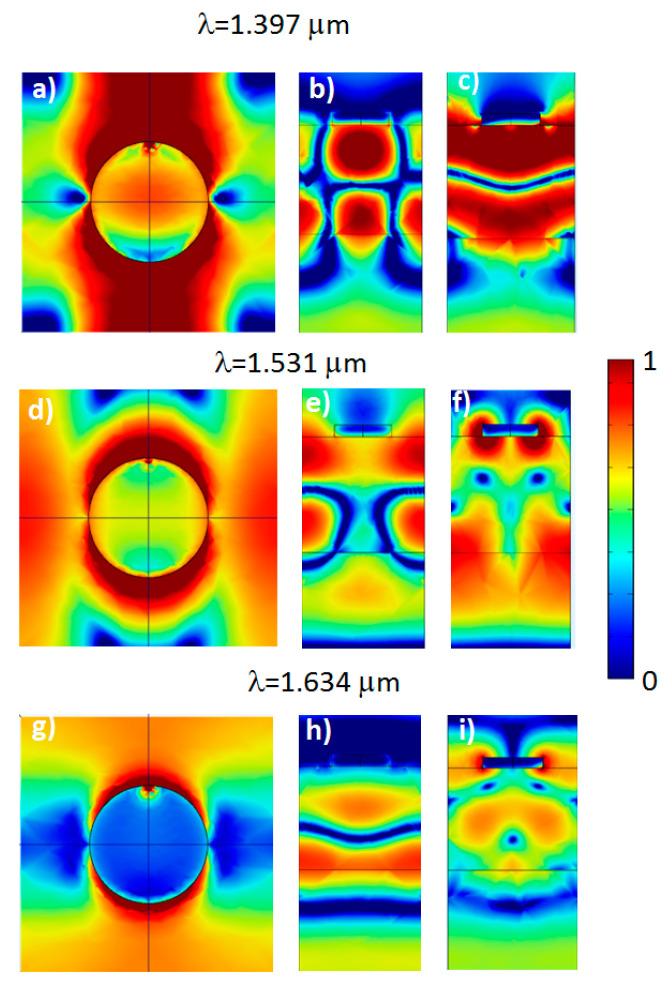
Near field distributions calculated for the model structure that is an analog of the experimental SOI structure with a square lattice of Al disks coupled to a Si waveguide layer (disk diameter is 250 nm and period is 540 nm) at λ = 1.397 µm (top panels), λ = 1.531 µm (center panels) and λ = 1.634 µm (bottom panels). XY—sections (**a**,**d**,**g**) are taken at the interface air/Si. XZ—sections (**b**,**e**,**h**) and YZ—sections (**c**,**f**,**i**) cross the center of Al disk. Point dipole oriented along *y*-direction is placed under the edge of Al disk at the depth 10 nm. The calculation was performed at radiation exit angle α = 0°. The color scale is shown in the right.

**Figure 7 nanomaterials-13-02422-f007:**
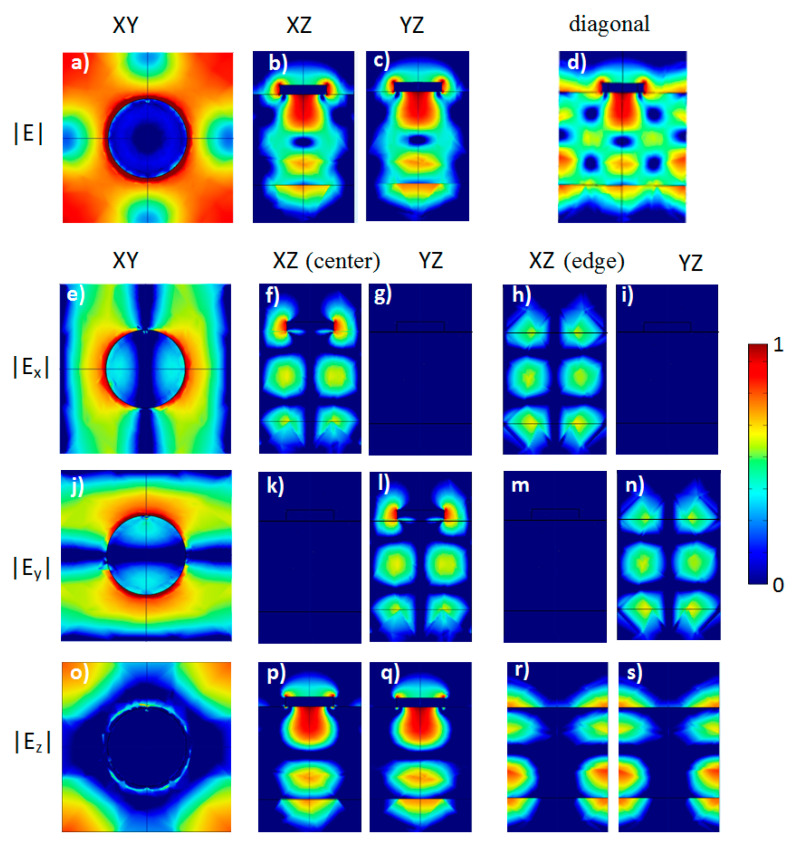
Near field distributions calculated at λ = 1.312 µm for the model structure that is an analog of the experimental SOI structure with a square lattice of Al disks coupled to a Si waveguide layer (*D* = 250 nm and *a* = 540 nm). XY—sections (**a**,**e**,**j**,**o**) are taken at the interface air/Si. XZ—sections (**b**,**f**,**k**,**p**) and YZ—sections (**c**,**g**,**l**,**q**) cross the center of Al disk, XZ—sections (**h**,**m**,**r**) and YZ—sections (**i**,**n**,**s**) are taken at the edge of the calculation cell, (**d**) diagonal section is passing through the center of Al disk. Point dipole oriented along *y*-direction is placed under the edge of Al disk at the depth 10 nm. The calculation was performed at radiation exit angle α = 0°. The color scale is shown in the right.

**Figure 8 nanomaterials-13-02422-f008:**
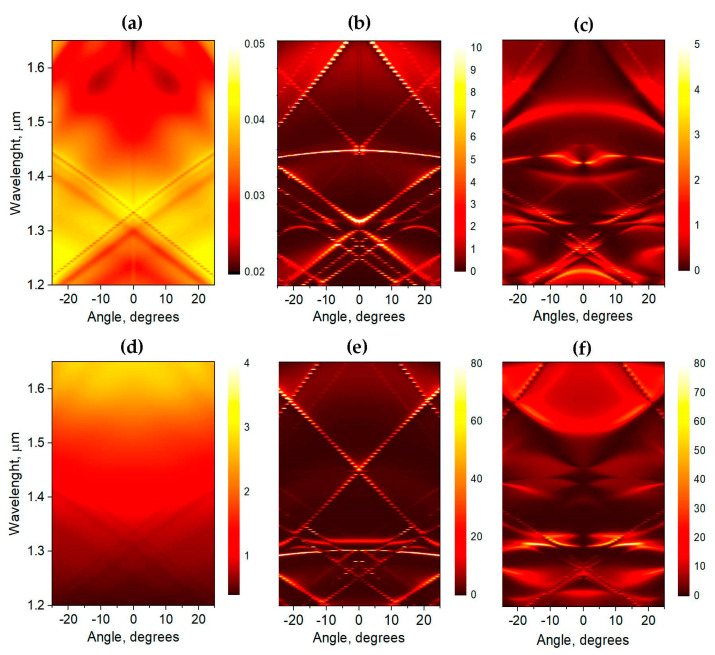
The *up*−emission intensity spectrum as a function of the radiation exit angle (a band diagram). Left panels (**a**,**d**) show the results of calculation for plasmonic structures with square lattice of Al disks on Si substrate. Center panels (**b**,**e**) show the results obtained for all-dielectric structures with square lattice of Ge disks on SOI structure. Right panels (**c**,**f**) show the case of metal–dielectric structures with square lattice of Al disks on SOI structure. In all cases the lattice parameters are the same, *a* = 540 nm and *D* = 250 nm. Top panels demonstrate the dispersion dependence of the *X*-dipole emission. Bottom panels show the dispersion dependence of the *Y*-dipole emission.

**Figure 9 nanomaterials-13-02422-f009:**
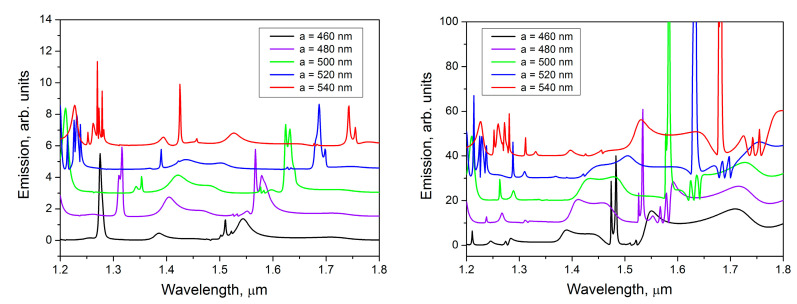
Calculated dependences of the emission intensity on the period of square lattice of Al disks with *X*-dipoles (**left panel**) and *Y*-dipoles (**right panel**) placed under the edge of Al disks in model SOI structure with 480 nm Si waveguide layer lying over a 3 μm SiO_2_ layer. The emission spectra are calculated at radiation exit angle α = 1°.

## Data Availability

Data are contained within the article.

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
