# Peer review of "Emission Enhancement of Ge/Si Quantum Dots in Hybrid Structures with Subwavelength Lattice of Al Nanodisks"

_nanomaterials, 2023, doi:10.3390/nano13172422_

Round 1
Reviewer 1 Report
The authors investigated resonance interaction between plasmonic and photonic modes in hybrid metal-dielectric structures. The study was systematically performed and matched well with simulation results, but there needs some improvement for clarity.
I recommend a publication after minor revision.
-
The resolution of Figure 1 and FIgure 4 are insufficient. I need to be improved to distinguish the text in the figure.
- There is no mention of Figures 4 and 9 in the manuscript. It is necessary to refer to the corresponding figure in the text to help the readers.
-
Some typographical errors need to be corrected.
In line 48, and 55, “gybrid” is considered a typo of “hybrid”
In line 329, there appears to be a duplication of "a = 480 nm." It seems that there might be a typographical error, and one of them should likely be revised as "a = 500 nm."
Author Response
Comment 1: The resolution of Figure 1 and FIgure 4 are insufficient. I need to be improved to distinguish the text in the figure.
Reply to comment 1: In new version of manuscript we have improved the resolution of all figures.
Comment 2: There is no mention of Figures 4 and 9 in the manuscript. It is necessary to refer to the corresponding figure in the text to help the readers.
Reply to comment 2: We have added the references to Figures 4 and 9.
Comment 3: Some typographical errors need to be corrected.
In line 48, and 55, “gybrid” is considered a typo of “hybrid”
In line 329, there appears to be a duplication of "a = 480 nm." It seems that there might be a typographical error, and one of them should likely be revised as "a = 500 nm".
Reply to comment 3: We have corrected these typos.
Reviewer 2 Report
In the paper, the effects of resonance interaction of plasmonic and photonic modes in hybrid metal-dielectric structures with square Al nanodisk lattices coupled to a Si waveguide layer were investigated by micro-photoluminescence spectroscopy. At optimal parameters of Al nanodisk lattices, almost one order increasing of PL intensity was obtained. The realization of high quality bound states in the continuum was confirmed by a comparative analysis of the experimental spectra and theoretical dispersion dependences. The results demonstrated the possibilty of the realization of bound states in the continuum in the metal-dielectric structures with square lattice of Al disks coupled to a Si waveguide layer.
The structure proposed in the manuscript can support two different types of bound states. Please clarify the significance of these two different types of bound states in the manuscript. In order to visualize the shift of the P2 peak position to the shorter wavelength region with the decrease of the lattice period, it is recommended that the positions of the P1 and P2 peaks should be indicated on the emission spectra of the other lattice periods (a=400nm-500nm) in the left panel of Fig. 2, The text in some figures is very unclear, such as Figures 1b, 4, and 8. The spectral lines in Figure 4 are also not very clear.
Author Response
Comment 1: The structure proposed in the manuscript can support two different types of bound states. Please clarify the significance of these two different types of bound states in the manuscript. In order to visualize the shift of the P2 peak position to the shorter wavelength region with the decrease of the lattice period, it is recommended that the positions of the P1 and P2 peaks should be indicated on the emission spectra of the other lattice periods (a=400nm-500nm) in the left panel of Fig. 2, The text in some figures is very unclear, such as Figures 1b, 4, and 8. The spectral lines in Figure 4 are also not very clear.
Reply to comment 1: In new version of manuscript we have added the words about the significance of two different types of bound states (see Introduction). We have indicated the positions of the P1 and P2 peaks by arrows in the left panel of Fig. 2. We have improved the resolution of all Figures.
Reviewer 3 Report
Comments on the manuscript
This work employs micro-photoluminescence spectroscopy to investigate the resonance interaction between plasmonic and photonic modes in hybrid metal-dielectric structures containing square aluminum nanodisk lattices coupled to a silicon waveguide layer. GeSi quantum dots embedded in the waveguide serve as radiation sources. The research observes narrow photoluminescence peaks within the quantum dot emission range and achieves enhanced photoluminescence intensity with optimal parameters for the nanodisk lattices. The results confirm the potential for realizing high-quality bound states in the continuum and supporting slow light. The article also discusses the application of photonic crystals and plasmonics in enhancing quantum dot emission and proposes a promising method for creating effective emitters with controlled plasmonic and photonic resonances using metal-dielectric structures.
In my assessment, the manuscript is interesting, and the simulation/experimental findings contribute to plasmonic-dielectric hybrid BICs, a topic of interest. Overall, the manuscript is methodologically robust, with well-support claims and conclusions. Consequently, this manuscript aligns with the scope of Nanomaterials, pending the resolution of a few minor concerns. Please find my suggestions and comments for the authors below.
1. The language is a little difficult to understand. For example, "There is already quite a lot of research in this direction [11-13], but so far no narrow high quality resonances typical of PhC have not been obtained." Narrow spectral linewidth resonance and high-Q factor convey equal concepts. It is struggling to figure out the meaning of "narrow high quality resonances typical of PhC"
2. Some claims lack literature support. For example, “The high quality factor is a intrinsic feature of special type of states, bound states in continuum. Usually, these states are observed in dielectric structures, and their realization in metallic systems is a very hard problem”. While this statement is accurate, it lacks corroborative substantiation. Nevertheless, recent progress in plasmonic bound states in the continuum (BICs) can support this assertion. Some recent advances in plasmonic BICs can support your claim. For example, studies on (1) plamonic BICs in anisotropic metasurfaces [Liang, Yao, et al. "Bound states in the continuum in anisotropic plasmonic metasurfaces." Nano Letters 20.9 (2020): 6351-6356.]. (2) chiral plasmonic BICs.
3. Some typos “Recently, the gybrid system with metallic grating coupled to a dielectric optical waveguide was investigated…” should be “hybrid”. Similar issues can be found throughout the manuscript. For example, “In the present work we study the effects of interaction of GeSi quantum dot emitters with gybrid plasmonic-photonic waveguide modes..”
4. On page 8, the interaction between photonic crystal structures and plasmonic modes is discussed, especially the cases where certain plasmonic modes closely coincide with photonic modes in k-space. Can the authors explain how this interaction alters the photonic crystal's band structure and emission properties in more detail? What mechanisms could explain the observed behavior where certain photonic modes are completely suppressed while others are repulsively shifted due to interaction?
readable
Author Response
Comment 1: The language is a little difficult to understand. For example, "There is already quite a lot of research in this direction [11-13], but so far no narrow high quality resonances typical of PhC have not been obtained. "Narrow spectral linewidth resonance and high-Q factor convey equal concepts. It is struggling to figure out the meaning of "narrow high quality resonances typical of PhC".
Reply to comment 1: We remove the word “narrow”.
Comment 2: Some claims lack literature support. For example, “The high quality factor is a intrinsic feature of special type of states, bound states in continuum. Usually, these states are observed in dielectric structures, and their realization in metallic systems is a very hard problem”. While this statement is accurate, it lacks corroborative substantiation. Nevertheless, recent progress in plasmonic bound states in the continuum (BICs) can support this assertion. Some recent advances in plasmonic BICs can support your claim. For example, studies on (1) plamonic BICs in anisotropic metasurfaces [Liang, Yao, et al. "Bound states in the continuum in anisotropic plasmonic metasurfaces." Nano Letters 20.9 (2020): 6351-6356.]. (2) chiral plasmonic BICs.
Reply to comment 2: We have added this reference.
Comment 3: Some typos “Recently, the gybrid system with metallic grating coupled to a dielectric optical waveguide was investigated…” should be “hybrid”. Similar issues can be found throughout the manuscript. For example, “In the present work we study the effects of interaction of GeSi quantum dot emitters with gybrid plasmonic-photonic waveguide modes”.
Reply to comment 3: We have corrected this misprint.